# Postoperative Radiotherapy for Endometrial Cancer in Elderly (≥80 Years) Patients: Oncologic Outcomes, Toxicity, and Validation of Prognostic Scores

**DOI:** 10.3390/cancers13246264

**Published:** 2021-12-14

**Authors:** Eva Meixner, Kristin Lang, Laila König, Elisabetta Sandrini, Jonathan W. Lischalk, Jürgen Debus, Juliane Hörner-Rieber

**Affiliations:** 1Department of Radiation Oncology, Heidelberg University Hospital, Im Neuenheimer Feld 400, 69120 Heidelberg, Germany; kristin.lang@med.uni-heidelberg.de (K.L.); laila.koenig@med.uni-heidelberg.de (L.K.); elisabetta.sandrini@med.uni-heidelberg.de (E.S.); juergen.debus@med.uni-heidelberg.de (J.D.); juliane.hoerner-rieber@med.uni-heidelberg.de (J.H.-R.); 2Heidelberg Institute of Radiation Oncology (HIRO), Im Neuenheimer Feld 400, 69120 Heidelberg, Germany; 3National Center for Tumor Diseases (NCT), Im Neuenheimer Feld 460, 69120 Heidelberg, Germany; 4Perlmutter Cancer Center, Department of Radiation Oncology, New York University Langone Health, New York, NY 10023, USA; jonathan.lischalk@nyulangone.org; 5Heidelberg Ion Therapy Center (HIT), Im Neuenheimer Feld 450, 69120 Heidelberg, Germany; 6German Cancer Research Center (DKFZ), Clinical Cooperation Unit Radiation Oncology, Im Neuenheimer Feld 280, 69120 Heidelberg, Germany

**Keywords:** geriatric care, vulnerability, Charlson Comorbidity Index (CCI), G8 geriatric screening score, uterine neoplasm, high-dose-rate brachytherapy, adjuvant therapy

## Abstract

**Simple Summary:**

As population ages, understanding of frailty in cancer patients becomes all the more important. Due to the rarity of elderly patients in randomized prospective trials, only limited data exist regarding safety and feasibility of postoperative radiotherapy for very elderly women of 80 years or older in a curative treatment intent. Further, geriatric assessments and prognostic scores for these women are not sufficiently validated. In a homogenous cohort of very elderly women with endometrial cancer, we confirmed that, despite older age, adjuvant radiotherapy can achieve excellent local control and overall survival with minimal high-grade toxicity. The geriatric G8 screening score was a highly applicable tool for prognostic evaluation of overall survival in our review.

**Abstract:**

Endometrial cancer is a common malignancy in elderly women that are more likely to suffer from limiting medical comorbidities. Given this narrower therapeutic ratio, we aimed to assess the oncologic outcomes and toxicity in the adjuvant setting. Out of a cohort of 975 women, seventy patients aged ≥ 80 years, treated with curative postoperative radiotherapy (RT) for endometrial cancer between 2005 and 2021, were identified. Outcomes were assessed using Kaplan–Meier-analysis and comorbidities using the Charlson Comorbidity Index and G8 geriatric score. The overall survival at 1-, 2- and 5-years was 94.4%, 82.6%, and 67.6%, respectively, with significant correlation to G8 score. At 1- and 5-years, the local control rates were 89.5% and 89.5% and distant control rates were 86.3% and 66.9%, respectively. Severe (≥grade 3) acute toxicity was rare with gastrointestinal (2.9%), genitourinary (1.4%), and vaginal disorders (1.4%). Univariate analysis significantly revealed inferior overall survival with lower RT dose, G8 score, hemoglobin levels and obesity, while higher grading, lymphangiosis, RT dose decrease and the omission of chemotherapy reduced distant control. Despite older age and additional comorbidities, elderly patients tolerated curative treatment well. The vast majority completed treatment as planned with very low rates of acute severe side-effects. RT offers durable local control; however, late distant failure remains an issue.

## 1. Introduction

Endometrial cancer (EC) represents the fourth most common cancer in women in the United States with a rising incidence and a mean age at diagnosis of 68 years [1,2]. Known risk factors for EC development include body mass index (BMI) and estrogen therapy [3,4,5]. The most common clinical presentation is postmenopausal bleeding which is typically evaluated by transvaginal ultrasound and subsequent endometrial sampling [6,7]. Pathological classification originally emerged from two classical phenotypes and includes Type I which is closely linked to aforementioned metabolic components, and Type 2 representing a higher risk cohort with poor differentiation and inferior clinical outcome [8]. Correlations of these two types with histologic patterns have been defined with the most common variants being endometrioid adenocarcinoma, serous and clear-cell carcinoma, and carcinosarcoma [9,10]. Staging follows the classification of the International Federation of Gynecology and Obstetrics (FIGO) Cancer Report [11] and corresponding TNM classification. 

Radical hysterectomy and bilateral salpingo-oophorectomy represent the cornerstone of curative therapy [11,12]. Ongoing discussions remain regarding the necessity and extent of surgical lymph node assessment and sentinel lymph node mapping in this disease site [13]. Adjuvant oncological management consists of a broad variety of radiotherapeutic and often times chemotherapy options. Risk-stratified recommendations are constantly updated, particularly with respect to the results of the Cancer Genome Atlas [14] and ESMO-ESGO-ESTRO consensus [15] which aim for a personalized molecular-based treatment approach. From a radiation standpoint, external beam radiotherapy (EBRT) is utilized in cases of locally advanced tumors or localized disease with demonstrated high risk factors. Adjuvant radiotherapy (RT) has been proven to significantly reduce locoregional relapse from 15.5% to 6%, as observed in the 15-year long-term PORTEC1-analysis [16]. Similarly, vaginal brachytherapy (BT) has been shown to achieve a comparable outcome to EBRT in PORTEC2-trial [17] for intermediate risk EC with concordant improvements in gastrointestinal toxicity. Ultimately, 5-year overall survival (OS) rates have been reported to be 81 to 96% for early-stages with a 2-year risk of local relapse at 3% following RT for intermediate-risk patients [16,18,19].

In the modern era, the use of patient-tailored adjuvant therapy aims to optimize oncologic outcome, however, similar to many published studies, it is often reserved for women with good clinical performance status. Elderly patients with multiple medical comorbidities may have worsened performance status, thus narrowing the therapeutic window for adjuvant treatment. Oncologic care for this patient population requires multidisciplinary management in effort to offer curative therapy while reducing toxicity and preserving quality of life. Nevertheless, despite EC being a disease of elderly women, oncogeriatric patients represent a minority of the patient population included in published studies [20,21,22]. As such, detailed research in this patient population is required to better characterize cancer and toxicity outcomes.

Comprehensive geriatric and comorbidity assessments currently exist to evaluate the oncogeriatric patient and include the Adult Comorbidity Evaluation 27 (ACE-27) [23], Charlson Comorbidity Index (CCI) [24], and G8 screening score [25]. Such assessments offer a standardized way to characterize the risk and feasibility of offering aggressive cancer treatment in this patient population. Unfortunately, most of these scales are tailored for perioperative mortality and surgical outcomes rather than RT [20,26]. Although, in the acute setting, RT can be much less toxic than surgery, the logistical burden of extended EBRT can be challenging and late toxicity requires better categorization. Ultimately, EC treatment for elderly patients is complex and consists of highly individual choices for which feasibility of optimal treatment must be evaluated.

Therefore, the aim of the current study is to evaluate oncologic outcome and treatment-related acute toxicity of postoperative curative RT in elderly patients diagnosed with EC, as well as analyze prognostic factors for improved identification of vulnerable patients. 

## 2. Materials and Methods

### 2.1. Patient and Treatment Characteristics 

Out of a sample of 975 women, who were treated with curative, postoperative RT following surgical resection for EC between March 2005 and June 2021 at our Department of Radiation Oncology of Heidelberg University Hospital, we included a subset of women aged ≥ 80 years. Primary focus was set by a threshold of 80 years due to the underrepresentation of this subgroup in prior and ongoing analyses and considering the growing elderly cancer population. The single-institutional retrospective analysis was approved by the local ethics committee (S-453/2021). Patient and oncologic data were individually assessed for each woman and included the following: Demographic, comorbidity, clinical gynecological exam, transvaginal ultrasound, computed tomography, magnetic resonance imaging and cysto- and rectoscopy, if available. Each patient was carefully classified according to FIGO staging system [11].

### 2.2. Radiotherapy and Oncological Treatment

Treatment recommendations were documented according to current national guidelines and patients’ individual status as discussed in an interdisciplinary tumor conference with input from gynecological oncology, radiation oncology, radiology, medical oncology, and pathology. All patients underwent upfront radical hysterectomy at a minimum, and were subsequently treated with adjuvant RT with curative intent. Stage-adapted surgical treatment was performed according to oncologic recommendations with the exception of cases where comorbidities precluded optimal surgical treatment. Of note, pelvic lymph node dissection was performed according to local institutional guideline recommendations, though in some cases, due to patient performance status or preference, this was deferred.

Radiotherapy was delivered in an adjuvant setting utilizing BT alone or BT in combination with EBRT. Radiotherapy modality was based on FIGO staging, although in some cases, due to poor performance status or patient refusal, EBRT was deferred. EBRT was delivered utilizing a 6–23 MV linear accelerator either as 3-dimensional conformal radiotherapy (3D-CRT) or intensity modulated radiation therapy (IMRT). Contouring of the clinical target volume (CTV) for EBRT was performed according to the updated institutional guidelines [27,28] and each patients’ individual risk for the pelvic region including the upper vagina, vaginal cuff, paravaginal and parametrial tissues. The nodal CTV further encompassed the following regions: the bifurcation of the aorta and presacral area as well as the common iliac, obturator and external and internal iliac. A margin of 0.5 to 1.5 cm was applied for the planning target volume (PTV) depending on the RT technique and organ motions. For EBRT, conventional dose fractionation was utilized delivered on a once daily schedule. Dose constraints for adjacent organs at risk (OARs) were set according to the Quantec recommendations [29]. All patients received a vaginal cuff high-dose-rate (HDR) BT, using Iridium-192 with intracavitary single or multichannel applicator according to American Brachytherapy Society Consensus Guidelines [30]. For further comparison, an equivalent dose in 2 Gy fractions (EQD2) was calculated for EBRT and HDR BT using the linear-quadratic model (1): EQD2Gy = fractional dose × number of fractions × (fractional dose + α/β)/(2Gy + α/β)(1)

Tumor dose summations were calculated with an α/β ratio of 10 assumed for the tumor.

### 2.3. Toxicity, Oncologic Follow-Up, and Prognostic Factors

Toxicity was classified according to Common Terminology Criteria for Adverse Events (CTCAE, version 5.0). Any new onset of symptoms within the start of RT up to 90 days was considered as acute toxicity, late-toxicity was defined by a time interval of >90 days during the follow-up period and assessed at first clinical examination after radiotherapy. Clinical oncologic outcome included the analysis of OS, local control (LC), and distant control (DC) and was defined as the time interval from the start of RT until death from any cause or at last contact, and until the first occurrence of local or distant recurrence. Local recurrence was considered as any tumor progression at the primary site or regional pelvic lymph nodes. DC was specified as development of metastatic lesions occurring outside the pelvis. For evaluation of treatment response, follow-up included any medical information from clinical examinations, referring physician notes and radiology.

For comorbidity and geriatric assessments, CCI [24] and G8 screening score [25] have been proven to have a high sensitivity in studies with cancer patients and prediction of age-adjusted comorbidities for estimation of 10-year survival rates [31]. Required factors, such as nutrition status, cognitive and depression scale, clinical performance, and comorbidities, were evaluated. Due to the retrospective nature of the study, reliable full detail data for G8 scoring were only available for a subset of women.

### 2.4. Statistical Analysis

Kaplan–Meier analysis and the log-rank test or Cox regression were utilized for the calculation of OS, LC, and DC during the follow-up period and further to compare survival curves and subgroups. A *p*-value of less than 0.05 was considered as statistically significant. Uni- and multivariate log-rank test and Cox-proportional hazard ratios (HR) with a 95% Confidence Interval (CI) were applied to analyze the influence of prognostic factors. Only significant variables from the univariate analysis were included to multivariate Cox models. Statistical analysis was performed with the statistical software IBM SPSS statistics (versions 25 and 28, Armonk, NY, USA).

## 3. Results

### 3.1. Patient and Tumor Characteristics

Out of 975 women treated with postoperative radiotherapy for EC, a total of 70 patients (7.2%) with a median age of 82 years (range: 80–95 years) were identified that met our inclusion criteria. From a performance status standpoint, the median Karnofsky score was 80 (60–100) with a median Charlson comorbidity index of 5 (4–9). The median G8 screening score was found to be 13 (6–16) and was assessed in a subset of 58 patients who had complete information available for analysis. The vast majority of patients in the overall cohort (≥80 years) and in the subgroup of patients, for whom the G8 screening was available, were diagnosed with FIGO stage I or II disease (overall cohort: *n* = 58, 82.9%; subgroup: *n* = 48, 82.8%). The predominant histologic subtype was endometrioid (*n* = 51, 72.9%), which is consistent with epidemiological data. For the definition of the histologic grading, serous carcinoma and carcinosarcoma were considered to be grading G3. Patient characteristics are detailed in Table 1.

### 3.2. Treatment Characteristics

Overall, the majority of patients underwent an oncologic surgery (*n* = 62, 88.6%) with the remaining patients undergoing a non-oncologic excision due to patient comorbidities or preference. There was a wide variety of lymph node assessment in our cohort with a relatively even distribution of patients not requiring a lymph node dissection, undergoing a lymph node dissection, or deferring a recommended dissection due to toxicity risk or patient preference. Of note, very few patients underwent adjuvant chemotherapy (*n* = 1, 1.43%) despite an indication for systemic therapy in over half of our cohort. This is a reflection of the limited performance status and median age of our patient population.

Postoperative radiation therapy was performed as indicated in the plurality of patients (*n* = 57, 81.4%). The median time from surgery to the start of radiation was 54 days with a median radiation treatment length of 25 days. Brachytherapy alone was performed in the majority of cases (*n* = 36, 51.4%), with the remaining receiving a combination of EBRT and BT. EBRT type was evenly distributed between IMRT (*n* = 16, 22.9%) and 3D-CRT (*n* = 18, 25.7%). For EBRT, a median dose of 45.0 Gy (19.8–56 Gy) in 11 to 28 once-daily fractions was employed. For women treated with BT only, a median dose of 22.0 Gy (5–22 Gy) in 1 to 4 fractions was delivered. In those who received BT as boost in combination with EBRT, a median boost dose of 10 Gy (10–25 Gy) in 2 to 5 fractions was used. Four patients (5.7%) did not complete radiation treatment as planned, three due to grade 3 acute toxicity, one due to refusal of any further adjuvant therapy. Treatment characteristics are detailed in Table 1 and Table 2.

### 3.3. Toxicity

The most common low-grade (≤grade 2) acute toxicity observed was as follows: gastrointestinal (*n* = 31, 44.3%), genitourinary (*n* = 25, 35.7%), vaginal mucositis (*n* = 17, 24.3%), and fatigue (*n* = 10, 14.3%) and significantly more often observed in patients with EBRT plus BT compared to BT alone (*p* = 0.013). Gastrointestinal toxicity (≤grade 2) in 31 patients included one of the following or a combination of diarrhea (*n* = 20), nausea (*n* = 6), abdominal pain (*n* = 5), and obstipation (*n* = 3), while genitourinary disorders were mainly dysuria (*n* = 14), urinary frequency/urgency (*n* = 12), incontinence (*n* = 6), and nycturia (*n* = 2). Extended para-aortic field RT was applied only once (*n* = 1, 81 years) due to more than twenty pathologically confirmed metastatic para-aortic lymph nodes, leading to grade 2 gastrointestinal toxicity without hospitalization but RT discontinuation for two days.

Overall, severe (≥grade 3) acute toxicity was rare and included the following rates: gastrointestinal (2.9%), genitourinary (1.4%), and vaginal disorders (1.4%). Higher grade acute toxicity occurred in two patients with grade 3 gastrointestinal symptoms (*n* = 2, 2.9%) of severe diarrhea with electrolyte abnormalities and one woman with grade 3 vaginal cuff dehiscence (*n* = 1, 1.4%) who had insufficient wound healing post hysterectomy. In each case of high-grade toxicity, this led to cessation of RT treatment early. One woman (1.4%) with urinary disorders and a grade 3 infection required intravenous antibiotics leading to urinary retention ultimately finished therapy without discontinuation with supportive care. Therapy interruption of one to five days was documented for 23 (32.9%) patients, leading to a premature termination of RT for four (5.7%) women. Inpatient treatment was necessary for 28 patients (40.0%) with a median duration of four (2–48) days. Median weight loss during RT was 0 (−10 to +7) kilogram. 

Concerning late-toxicity, fatigue (≤grade 2) was the most common side effect, occurring in nine patients (12.9%). Only two patients experienced a new onset of gastrointestinal toxicity (2.9%) in the form of low-grade (≤grade 2) obstipation (*n* = 1) and nausea plus diarrhea (*n* = 1), while three patients suffered from new and increasing genitourinary (4.3%) side effects with worsening incontinence (*n* = 2) and new dysuria (*n* = 1).

### 3.4. Oncologic Outcomes

Oncologic outcomes for the overall group of 70 patients with a median follow-up time of 4.9 months (0–128.0 months) is shown in Table 3. Kaplan–Meier curves for oncologic outcomes are displayed in Figure 1. The 1-, 2- and 5-year OS rates were 94.4%, 82.6%, und 67.6%, respectively (Figure 1A). Local failures were detected in three women (4.3%) within the first year after RT, with a median time to relapse of 9.6 months (4.5–10.2 months). This resulted in 1-, 2-, and 5-year LC rates of each 89.5% (Figure 1B). During the follow-up period, a total of nine patients were diagnosed with distant metastases after a median of 12.2 months following RT (2.6–49.3) and with the following organ distribution: peritoneal (*n* = 4), lymphatic outside pelvis (*n* = 4), pulmonary (*n* = 2), bone (*n* = 2), hepatic (*n* = 1), spleen (*n* = 1), brain (*n* = 1), skin (*n* = 1), pleural (*n* = 1) lesions. DC rates were of 86.3%, 73.6% and 66.9% (Figure 1C) after 1-, 2- and 5-years, respectively.

### 3.5. Prognostic Factor Analysis 

Univariate analysis of prognostic factors influencing oncologic outcomes was performed in detail (Table 4). None of the assessed clinical or tumor-related factors were found to be associated with local control outcome. Univariate analysis revealed lower histologic tumor grading (Hazard Ratio (HR) 34.098, (Confidence Interval (CI): 0.82–1411.8), *p* = 0.001) and absence of lymph vascular involvement (HR 5.581 (CI: 0.979–31.829), *p* = 0.031) were significantly associated with lower rates of distant metastases. Additionally, treatment with an EQD2 below 60 Gy (HR 8.969 (CI: 1.766–45.543), *p* = 0.008) and the omission of adjuvant chemotherapy was highly associated with inferior DC (HR 2.236 (CI: 0.925–5.405), *p* < 0.0001). Multivariate analysis could not identify any statistically significant associations for either LC or DC.

The extent of the tumor according to the TNM classification was a reliable prognostic factor for inferior OS in locally advanced tumors (HR 34.490 (CI: 2.158–551.55), *p* = 0.012). Furthermore, the histological tumor subtype was significantly associated with the outcome of OS (HR 3.400 (CI: 1.427–8.100), *p* = 0.006) with endometroid adenocarcinoma being the most favorable and carcinosarcoma being the least favorable histologic subtype. A BMI of 28 or higher (HR 1.102 (CI: 0.989–1.228), *p* = 0.033) and anemia with a hemoglobin level below 12 (HR 0.155 (CI: 0.019–1.267), *p* = 0.046) were significantly associated with an inferior OS on univariate analysis. Utilization of an equivalent dose in 2-Gy fractions (EQD2 dose) for a α/β = 10 of at least 60 Gy showed superior OS outcome (HR 33.062 (CI: 3.618–302.1), *p* = 0.002). For frailty assessment, the G8 rating (HR 0.666 (CI: 0.503–0.883), *p* = 0.005) was identified as a significant factor for OS prediction, whereas CCI score was not reliable for this patient cohort. Multivariate analysis was not significant for OS for any of the evaluated factors. 

## 4. Discussion

Endometrial cancer is a common gynecologic malignancy of elderly women with a growing spectrum of individualized therapeutic approaches due to constantly updated molecular and genetic research. Elderly patients with multiple medical comorbidities represent a large cohort of patients treated in the clinic, but are often times underrepresented in published randomized trials [32]. Therapeutic recommendations based strictly on age stratification are insufficient in differentiating fit versus frail patients, particularly as life expectancy continues to improve. Theoretically, from a technical standpoint, there is almost no limit to aggressive radiotherapeutic approaches making patient selection in a multidisciplinary setting critical to determine a personalized treatment approach based not only on patient age but medical comorbidities, performance status, and patient preference.

Certainly, age alone has been described as a risk factor for inferior outcomes in the historical literature [33,34]. Alektiar et al. [34] reported 5-year OS of EC patients < 70 years (*n* = 321) was significantly higher than in women age ≥ 70 years (*n* = 84), 92% vs. 80% (*p* = 0.006), respectively. In fact, a comparable detriment in oncologic outcome was observed for 5-year LC rates in patients age greater than 70 years (89% vs. 97%, *p* = 0.02). A patient cohort of 53 women age ≥ 75 years treated with postoperative RT for EC was analyzed by Fiorica et al. [35] and they found a 5-year OS of 72.3% and a 5-year LC of 82.7%. In the present study, a larger group of 70 women with different inclusion criteria, all aged 80 years or older, were analyzed and showed an estimated 5-year LC rate of 89.5%, comparable to the aforementioned data. In contrast, our data showed an inferior 5-year OS rate of only 67.7%, which is likely due to the comparable older age of the present cohort (median age of 82 years).

The omission of postoperative RT has been reported more often in older patients and is consistent with observed inferior LC [36]. Our study could not prove a significant correlation of outcome to the omission of indicated EBRT in a small number of 13 patients. However, despite the deferral of EBRT, BT alone was performed in this same subset. This was thought to be an optimal approach given the short overall treatment time and mitigation of treatment-related toxicity. This method has been previously reported in the literature to be a successful management strategy for women in poor general health [37]. 

Interestingly, the omission of chemotherapy (in warranted cases) led to an inferior distant control rate, which was notable in the oncologic outcomes of the present study. Indeed, in managing patients with a very narrow therapeutic window, the feasibility of toxic chemotherapy is restricted, which reflects the limited performance status and median age of our patient population. Thus, modification and risk-stratification for a less toxic systemic therapy regimes is a critical topic for future research [38]. 

Treatment-related toxicity is an important aspect of the analysis of any treatment regimen offered to high-risk patient population. Fiorica et al. [35] reported an elevated occurrence of low-grade gastrointestinal toxicities of 73.6%, but without occurrence of any severe toxicity. In comparison, our results showed a less notable low-grade toxicity rate of 44.3%, however, two patients developed grade 3 radiation-induced toxicity. Of note, no patients experienced any grade 4 or higher toxicity. Genitourinary, gastrointestinal, and vaginal side effects are well known in the treatment of gynecological cancers, but in our cohort, overall rates of severe toxicity were limited in these realms. Nonetheless, even in the presence of only so-defined ‘low’-grade toxicity, the rate of unplanned hospitalization in our study was high (40.0%) and prior studies of gastrointestinal cancer patients over 65 years have shown even higher rates up to 53% for a different RT region [39]. This is most likely reflecting the overall higher risk of therapy-induced impairment of quality of life and physical or psychological status in the very elderly due to frailty and clinical comorbidities [40].

It is worth pointing out that fatigue was a common toxicity observed by our elderly cohort. As described by Giacalone et al. [41], radiation-related fatigue can be multifactorial in nature, particularly in the elderly. Of note, our study also showed poorer overall survival in the presence of anemia, a common cause of fatigue. Thus, especially in older patients, special focus should be placed on the management of fatigue during RT. Duska et al. [21] advocate for a comprehensive offer of standard treatment approaches also to the elderly, but with thoughtful consideration and implementation of toxicity-sparing techniques, such as image-guided IMRT has proven in the treatment of pelvic gynecological malignancies [42]. Furthermore, from a radiation oncologist view, there has been great RT technique development over the assessed time of the last 15 years. The application of the standard technique has today changed from 3D-RT into widespread advanced IMRT, that has been proven to improve bone marrow sparing and more conformal target coverage and thus to be associated with lower gastrointestinal and hematologic toxicity than 3D-RT without diminishing the oncologic outcome [43]. The use of above-mentioned BT or the application of smaller pelvic RT volumes can be options to spare toxicity [44] and are even more attractive in the treatment of the very elderly, but those de-escalation strategies should be used after reasonable case evaluation only. For irradiation of lymphatic tissue, IMRT should preferably be used and is the standard of care, especially when extended para-aortic field RT is considered to be included into the target volume [45].

Data reliability of screening scores shows a promising correlation for the identification of frailty in heterogeneous cancer patient cohorts [46,47]. However, these data are sparse in the patient population that we assessed in the present study, that is, very elderly patients treated with radiotherapy. Fiorica et al. [35] have established an ACE-27 index to be significantly associated with OS for 53 EC patients of ≥75 years. Prominently, in our results, the G8 screening score was significantly predictive for OS. A comprehensive view of the geriatric patient can be ascertained using a multidimensional assessment that not only evaluates the patient’s physical health with respect to medical comorbidities, but also assesses psychosocial status and functional abilities. A subjective evaluation can be very time-consuming to the busy provider, thus screening scores including the G8 seem to be a feasible and standardized approach which can achieve optimal results. 

Indeed, special emphasis must be placed on reduction of treatment-related toxicity and preservation of quality of life in the elderly population. However, we advocate for strong consideration of curative treatment in this patient population which clearly has demonstrated negative oncologic results when treatment is deferred given excessive concern regarding patient age. Despite patients in our cohort all being age 80 years or older, overall clinical outcomes were quite good and relatively in line with their younger peers. Ultimately, multidisciplinary supportive care from a holistic standpoint is critical to optimize clinical outcomes in these patients [48,49]. In this context, future research should not rigorously exclude this underrepresented, growing subgroup, but has to re-define clinical endpoints and maybe even toxicity classifications. Further focus on studies dedicated to the very elderly is mandatory to provide more evidence for clinicians, taking into consideration that a more age-adjusted treatment approach and assessment of quality of life, changes the way how cancer therapy is implemented in clinical routine.

Major limitations of the present study include the retrospective nature of the review, which led to exclusion of patients with incomplete data for G8 rating. Subgroup analyses must therefore be interpreted cautiously and further prospective contributions to the sparse literature are greatly needed. An additional limitation is the short follow-up, albeit a consequence of the patients’ age and medical comorbidities; long-term follow-up can thus be challenging, especially scheduling outpatient follow-up appointments. Nevertheless, our cohort is one of the larger single-centers reporting outcomes and acute toxicities for a homogenous patient cohort of very elderly women age ≥ 80 years diagnosed with EC and treated adjuvant with radiotherapy with curative intent.

## 5. Conclusions

Postoperative radiotherapy in elderly patients aged 80 years or older diagnosed with EC achieves excellent LC and OS with minimal high-grade toxicity. However, distant failure with longer follow-up remains high due to the lack of safe systemic therapy options. The geriatric G8 screening score was highly applicable for OS prognostic evaluation in our cohort. We strongly advocate for multidisciplinary assessment of the individualized oncogeriatric patient to provide safe curative postoperative treatment for those diagnosed with EC.

## Figures and Tables

**Figure 1 cancers-13-06264-f001:**
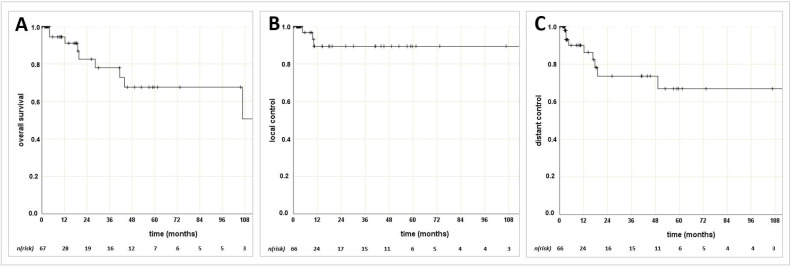
Kaplan–Meier curves for (**A**) overall survival, (**B**) local control, (**C**) distant control, *n* (*risk*): number at risk.

**Table 1 cancers-13-06264-t001:** Patient characteristics.

Characteristics	Values
Median age	82 (range: 80–95) years
Median Karnofsky performance score	80 (60–100) %
Median Charlson Comorbidity Index	5 (4–9)
Median G8 screening score	13 (6–16) *
Median BMI	26.5 (20.5–45.7)
Median hemoglobin level	11.9 (8.9–14.0) g/dL
FIGO stage	
1/2	58 (82.9%)
3/4	12 (17.1%)
Surgical procedure	
oncological procedure	62 (88.6%)
non-oncologic excision only	8 (11.4%)
Adjuvant chemotherapy	
not indicated	30 (42.9%)
performed	1 (1.43%)
indicated, but not feasible	39 (55.7%)
Lymph node dissection	
not indicated	20 (28.6%)
performed	25 (35.7%)
indicated, but not feasible	25 (35.7%)
Postoperative radiotherapy	
performed as indicated	57 (81.4%)
EBRT indicated, but not feasible	13 (18.6%)
Histological subtype	
endometrioid	51 (72.9%)
serous	13 (18.6%)
carcinosarcoma	6 (8.6%)

* assessed in 58 patients only. BMI: body-mass index, EBRT: external beam radiotherapy, FIGO: International Federation of Obstetrics and Gynecology.

**Table 2 cancers-13-06264-t002:** Treatment characteristics.

Characteristics	Values
Median time from surgery to start of RT	54 (range: 6–219) days
Median treatment time	25 (1–67) days
Median days of RT discontinuation	0 (0–5) days
Premature cessation of RT	4 (5.7%)
Median total dose in EQD2 (α/β = 10)	28.42 (6.25–77.98) Gy
RT technique	
IMRT (EBRT + BT)	16 (22.9%)
3D-CRT (EBRT + BT)	18 (25.7%)
BT only	36 (51.4%)

3D-CRT: three-dimensional conformal radiotherapy, BT: brachytherapy, EQD2: equivalent dose in 2 Gy fractions, IMRT: intensity-modulated radiotherapy, RT: radiotherapy.

**Table 3 cancers-13-06264-t003:** Outcome with Kaplan–Meier estimates.

Rates	Overall Survival	Local Control	Distant Control
1-year	94.4%	89.5%	86.3%
2-year	82.6%	89.5%	73.6%
5-year	67.6%	89.5%	66.9%

**Table 4 cancers-13-06264-t004:** Univariate analyses.

Characteristics	Overall Survival		Local Control		Distant Control	
	HR 95%CI	*p*	HR 95%CI	*p*	HR 95%CI	*p*
Age	0.937 (0.732–1.200)	0.608	1.005 (0.734–1.375)	0.976	0.668 (0.440–1.015)	0.059
Body mass index BMI < 28 vs. ≥28	1.102 (0.989–1.228)	**0.033**	1.112 (0.778–1.589)	0.208	0.936 (0.696–1.259)	0.137
Charlson Comorbidity Index	1.197 (0.706–2.030)	0.505	1.401 (0.648–3.026)	0.391	0.657 (0.335–1.288)	0.221
G8 Screening Score	0.666 (0.503–0.883)	**0.005**	1.438 (0.700–2.955)	0.323	0.936 (0.666–1.317)	0.705
Karnofsky performance score ≤80 vs. ≥90	1.215 (0.245–6.030)	0.811	0.034 (0–4223.770)	0.572	3.049 (0.816–11.397)	0.081
Number of medications	0.898 (0.690–1.167)	0.420	0.970 (0.547–1.723)	0.918	0.931 (0.710–1.220)	0.603
Hemoglobin level ≤12 vs. >12 g/dL	0.155 (0.019–1.267)	**0.046**	47.435 (0–5,464,301)	0.516	0.522 (0.124–2.195)	0.375
Histologic grading G1 vs. G2 vs. G3	2.412 (0.749–7.767)	0.140	1.274 (0.260–6.256)	0.765	34.098 (0.82–1411.8)	**0.001**
FIGO stage 1/2 vs. 3/4	0.993 (0.122–8.089)	0.995	0.042 (0–335,213.6)	0.695	1.881 (0.388–9.122)	0.433
TNM T1/2 vs. T3/4	34.49 (2.158–551.6)	**0.012**	0.031 (0–1604.964)	0.529	8.762 (0.792–96.963)	0.077
Nodal stage N0 vs. N+	0.993 (0.122–8.089)	0.995	0.042 (0–335,213.6)	0.695	2.110 (0.434–10.251)	0.354
Tumor type endometrioid vs. serous vs. carcinosarcoma	3.400 (1.427–8.100)	**0.006**	0.052 (0–294.620)	0.503	1.770 (0.638–4.914)	0.273
Lymph vascular involvement L1 vs. L0	0.470 (0.055–4.033)	0.491	1.149 (0.104–12.68)	0.910	5.581 (0.979–31.829)	**0.031**
EQD2 ≥ 60 Gy (α/β = 10)	33.062 (3.618–302)	**0.002**	0.038 (0–32,130.29)	0.639	8.969 (1.766–45.543)	**0.008**
Omission of indicated therapy						
Oncological surgery	0.043 (0–2466.448)	0.574	0.044 (0–2,913,770)	0.733	0.040 (0–277.695)	0.476
Lymph node dissection	1.094 (0.459–2.609)	0.839	0.616 (0.134–2.836)	0.534	1.636 (0.665–4.026)	0.284
Chemotherapy	1.122 (0.572–2.202)	0.738	0.733 (0.221–2.435)	0.613	2.236 (0.925–5.405)	**<0.0001**
EBRT	1.013 (0.199–5.164)	0.987	0.039 (0–44,883.5)	0.649	0.600 (0.074–4.873)	0.633

BMI: body-mass index, EBRT: external beam radiotherapy, EQD2: equivalent dose in 2 Gy fractions, FIGO: International Federation of Obstetrics and Gynecology, CI: confidence interval, DC: distant control, HR: hazard ratio, LC: local control, OS: overall survival, RT: radiotherapy. For continuous variables, the HRs represents a relative increase in risk according to unit change. A *p*-value of <0.05 (bold in the table) was considered statistically significant.

## Data Availability

The data presented in this study were obtained from local databases of the Cancer Registry of the National Center for Tumor Diseases (NCT). The data are not publicly available due to Local Ethics Committee rules.

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
