# Peer review of "Postoperative Radiotherapy for Endometrial Cancer in Elderly (≥80 Years) Patients: Oncologic Outcomes, Toxicity, and Validation of Prognostic Scores"

_cancers, 2021, doi:10.3390/cancers13246264_

Round 1

Reviewer 1 Report

This manuscript by Meixner and co-authors is centred on the management an outcome of postoperative radiotherapy for endometrial cancer in elderly patients – an area of limited focused investigation to date. I enjoyed reading this paper and have a number of comments to report, as detailed below:

Introduction: The introduction is comprehensive, and suitably makes the point that despite the fact that endometrial cancer affects older women, oncogeriatric patients remain underrepresented in both studies and the wider literature.

Materials & Methods: These are described in adequate detail, including patient selection criteria and radiotherapeutic management. I wonder whether the authors could include a clearer justification for why they defined their age inclusion as 80 years given that other studies in the field have opted for a lower age bracket cutoff. The statistics are succinctly described but entirely appropriate. It would be useful to include the patient sample size early on in this section rather than mentioning it first in the results, with greater clarity regarding the subset of women meeting the requisite criteria for analysis. Given that this is a single centre-based analysis, the numbers are rather small but nevertheless acceptable given the stringent inclusion criteria based on age.

Results: The results are presented clearly and the analysis has been thorough. However, Table 4’s Grading (G1 vs. G2 vs. G3) is still a little unclear to me. Is this referring to tumour grade? If so, the authors should detail their grade allocation (with relevance to non-endometrioid subtypes being assumed to be high grade de facto). Also, is there any reason why "malignant mixed Müllerian tumor" is used specifically as a term in this table alone?

Discussion: The discussion is succinct, cogent and well written. Importantly, it acknowledges the limitations of the study. I wonder whether stronger emphasis can be made in terms of what this means translationally for clinical management and, perhaps more importantly, how age should focus of future research endeavours (trial-based and otherwise).

The manuscript requires very minor syntactical review but is otherwise fine and addresses an area of interest in oncological management.

Author Response

This manuscript by Meixner and co-authors is centred on the management an outcome of postoperative radiotherapy for endometrial cancer in elderly patients – an area of limited focused investigation to date. I enjoyed reading this paper and have a number of comments to report, as detailed below:                    Introduction: The introduction is comprehensive, and suitably makes the point that despite the fact that endometrial cancer affects older women, oncogeriatric patients remain underrepresented in both studies and the wider literature.  Materials & Methods: These are described in adequate detail, including patient selection criteria and radiotherapeutic management. I wonder whether the authors could include a clearer justification for why they defined their age inclusion as 80 years given that other studies in the field have opted for a lower age bracket cutoff. The statistics are succinctly described but entirely appropriate. It would be useful to include the patient sample size early on in this section rather than mentioning it first in the results, with greater clarity regarding the subset of women meeting the requisite criteria for analysis. Given that this is a single centre-based analysis, the numbers are rather small but nevertheless acceptable given the stringent inclusion criteria based on age.

>>>>> Thank you for very much for your comments and helpful suggestions you have made to enhance our manuscript. We have thoroughly revised it and adapted it according to the comments made. We have included our reasons for focussing on this subgroup of patients in the Materials & Methods section. As the population is aging and the number of older cancer patients is increasing, we aimed to contribute more evidence for the very elderly that, we assume, might be at an even higher risk of omission of therapy for reasons of comorbidity or social aspects.

Results: The results are presented clearly and the analysis has been thorough. However, Table 4’s Grading (G1 vs. G2 vs. G3) is still a little unclear to me. Is this referring to tumour grade? If so, the authors should detail their grade allocation (with relevance to non-endometrioid subtypes being assumed to be high grade de facto). Also, is there any reason why "malignant mixed Müllerian tumor" is used specifically as a term in this table alone?

>>>>> We do totally agree and further specified the term “grading” to “histologic grading” and added more detail as pathological reports considered serous carcinoma and carcinosarcoma as grading G3. All changes can be viewed by “track changing function” in Word. The term “malignant mixed Müllerian tumor” was used synonymously with carcinosarcoma in the text, we changed the term to avoid any misunderstanding.

Discussion: The discussion is succinct, cogent and well written. Importantly, it acknowledges the limitations of the study. I wonder whether stronger emphasis can be made in terms of what this means translationally for clinical management and, perhaps more importantly, how age should focus of future research endeavours (trial-based and otherwise).

>>>>> We added a statement to the discussion section to emphasize the need for further clinical studies dedicated to the very elderly, taking into consideration that clinical endpoints should reflect the age-dependent treatment aims. 

The manuscript requires very minor syntactical review but is otherwise fine and addresses an area of interest in oncological management.

Reviewer 2 Report

The authors conducted a retrospective study dedicated to the outcomes of post-surgical radiotherapy treatment of endometrial cancer in women over the age of 80.

Despite the limitations, recognized by the authors, due to the fact that it is a retrospective study, the results obtained are very interesting and certainly useful for all clinicians who are dealing with elderly patients affected by this malignant tumor.

The only weak point of the work is the retrospective collection of data from patients treated over 15 years. For this reason, the authors could perhaps say whether the treatment modalities of patients have significantly changed over the 15 years. Beyond that they could also clarify how many of the 58 patients for whom the G8 screening score was available had a FIGO 1/2 stage.

Author Response

The authors conducted a retrospective study dedicated to the outcomes of post-surgical radiotherapy treatment of endometrial cancer in women over the age of 80. Despite the limitations, recognized by the authors, due to the fact that it is a retrospective study, the results obtained are very interesting and certainly useful for all clinicians who are dealing with elderly patients affected by this malignant tumor.

The only weak point of the work is the retrospective collection of data from patients treated over 15 years. For this reason, the authors could perhaps say whether the treatment modalities of patients have significantly changed over the 15 years.

>>>>> Thank you for very much for your comments and helpful suggestions you have made to enhance our manuscript. We have thoroughly revised it and adapted it according to the comments made. We added a section about the RT technique development (from 3D-RT to widespread advanced IMRT) to the discussion section, focussing on important improvements for lower toxicity. All changes can be viewed by “track changing function” in Word.

Beyond that they could also clarify how many of the 58 patients for whom the G8 screening score was available had a FIGO 1/2 stage.

>>>>> We added detailed data for the subgroup of patients for whom the G8 screening was available: FIGO 1/2 stage was documented in 48 patients (82.8%), while 10 women (17.2%) had a FIGO stage 3/4 in the subgroup.

Reviewer 3 Report

The review reads well. The authors have described the outcomes of different radiotherapy techniques such as IMRT, EBRT and 3D-CRT in 975 women. The reports are well presented with statistical numbers. However, the authors should report comparison of different RTs, describe them in the discussion or conclusion. Which RT is better and favorable in the majority of cases? The side effects must be emphasized and elaborate more. 

Author Response

The review reads well. The authors have described the outcomes of different radiotherapy techniques such as IMRT, EBRT and 3D-CRT in 975 women. The reports are well presented with statistical numbers. However, the authors should report comparison of different RTs, describe them in the discussion or conclusion. Which RT is better and favorable in the majority of cases?

>>>>> Thank you for very much for your comments and helpful suggestions you have made to enhance our manuscript. We have thoroughly revised it and adapted it according to the comments made. After listing advantages concerning shorter overall treatment times and lower toxicity of brachtherapy in the Discussion, we added more detail of the advantages of advanced IMRT, leading to lower gastrointestinal and hematologic toxicity. All changes can be viewed by “track changing function” in Word.

The side effects must be emphasized and elaborate more. 

>>>>> We extended the Results section with inclusion of a more specific listing of acute-toxicities and included late-toxicity as assessed at the first follow-up appointment > 90 days. We further focused on the greater risk of hospitalization and therapy-induced impairment of quality of life and physical or psychological status in the very elder even in the presence of so called and defined ‘low’-grade toxicity.